# Applications of Electrospun Nanofibers with Antioxidant Properties: A Review

**DOI:** 10.3390/nano10010175

**Published:** 2020-01-20

**Authors:** Ariel Vilchez, Francisca Acevedo, Mara Cea, Michael Seeger, Rodrigo Navia

**Affiliations:** 1Doctoral Program in Sciences of Natural Resources, Universidad de La Frontera, Casilla 54-D, Temuco, Chile; a.vilchez01@ufromail.cl; 2Department of Basic Sciences, Faculty of Medicine, Universidad de La Frontera, Casilla 54-D, Temuco, Chile; francisca.acevedo@ufrontera.cl; 3Scientific and Technological Bioresource Nucleus, BIOREN, Universidad de La Frontera, Casilla 54-D, Temuco, Chile; mara.cea@ufrontera.cl; 4Department of Chemical Engineering, Faculty of Engineering and Sciences, Universidad de La Frontera, Casilla 54-D, Temuco, Chile; 5Laboratorio de Microbiología Molecular y Biotecnología Ambiental, Departamento de Química & Centro de Biotecnología (CBDAL), Universidad Técnica Federico Santa María, Avenida España 1680, Valparaíso 2340000, Chile; michael.seeger@usm.cl; 6Centre for Biotechnology and Bioengineering (CeBiB), Faculty of Engineering and Sciences, Universidad de La Frontera, Casilla 54-D, Temuco, Chile

**Keywords:** nanofibers, antioxidant activity, tissue engineering, food

## Abstract

Antioxidants can be encapsulated to enhance their solubility or bioavailability or to protect them from external factors. Electrospinning has proven to be an excellent option for applications in nanotechnology, as electrospun nanofibers can provide the necessary environment for antioxidant encapsulation. Forty-nine papers related to antioxidants loaded onto electrospun nanofibers were categorized and reviewed to identify applications and new trends. Medical and food fields were commonly proposed for the newly obtained composites. Among the polymers used as a matrix for the electrospinning process, synthetic poly (lactic acid) and polycaprolactone were the most widely used. In addition, natural compounds and extracts were identified as antioxidants that help to inhibit free radical and oxidative damage in tissues and foods. The most recurrent active compounds used were tannic acid (polyphenol), quercetin (flavonoid), curcumin (polyphenol), and vitamin B_6_ (pyridoxine). The incorporation of active compounds in nanofibers often improves their bioavailability, giving them increased stability, changing the mechanical properties of polymers, enhancing nanofiber biocompatibility, and offering novel properties for the required field. Although most of the polymers used were synthetic, natural polymers such as silk fibroin, chitosan, cellulose, pullulan, polyhydroxybutyrate, and zein have proven to be proper matrices for this purpose.

## 1. Introduction

Antioxidants can inhibit the oxidation of other compounds and prevent free radical formation, maintaining a dynamic equilibrium between free radical production and antioxidant levels, thus preventing oxidative stress in biological systems [1]. Vitamin C is a common example of an antioxidant present in foods. Foods such as berries, paprika, and oregano leaves, among others, are well recognized for their antioxidant abilities [2]. In the present review, most of the compounds are phenols and polyphenols, including monoterpenoids, hydroxycinnamic acids and flavonoids, but pigments, oleoresin, vitamins, carotenoids, proteins, and several natural extracts are also present. The action mechanisms of antioxidants in general include sequestration or the inhibition of free radicals, the chelation of metallic ions, and the prevention of protein modification, lipid peroxidation, or oxidative damage [3]. Enzymatic antioxidants such as superoxide dismutase or glutathione peroxidase act by decomposing and removing free radicals. All antioxidants presented in this article are not endogenous, belonging to a nonenzymatic antioxidant classification. They mainly act by scavenging reactive oxygen species, interrupting free radical reactions, and preventing the formation of oxidants through the donation of a hydrogen atom to free radicals, resulting in stabilized radicals. Flavonoids such as quercetin also protect DNA against oxidative damage, forming complexes with chelating metal ions and preventing the formation of reactive oxygen species (ROS) [2,4]. In medicine, the proper care and healing of infected and noninfected wounds can be enhanced through the action of antioxidant compounds such as free radical scavenging agents. This can be observed in the increased number of proliferating cells and blood vessels [5,6,7]. However, the bioavailability and bioaccessibility of antioxidants are limited by external factors such as pH, temperature, or light exposure, which can cause these compounds to degrade. Consequently, antioxidants have been encapsulated to enhance their availability, thereby maintaining their properties. Due to their varied mechanisms, the antioxidant activity of compounds is determined by different methods, the most common (in the reviewed articles) being the use of UV-VIS spectrophotometry to measure the final absorbance. The 2,2-diphenyl-1-picrylhydrazyl (DPPH) assay has been widely used. This free radical reacts in the presence of antioxidant compounds, accepting a hydrogen atom; thus, the absorbance of the violet solution decreases as a function of time and is measured at 515–520 nm. Similarly, the 2,2′-azino-bis(3-ethylbenzothiazoline-6-sulfonic acid) (ABTS) assay determines the capability of antioxidants to scavenge ABTS generated in aqueous medium in comparison to Trolox. The ferrous chelating assay determines the ability of antioxidants to chelate ferrous ions in an iron (II)–ferrozine complex solution, decreasing its absorbance. The phosphomolybdenum assay allows for assessing the antioxidant activity at 695 nm due to the reduction of Mo (VI) to Mo (V), and the results are commonly expressed in terms of equivalents of ascorbic or gallic acids. Fluorescence can also indicate antioxidant activity, and it is used for live cells to measure ROS. It needs an ROS-sensible indicator such as 2′,7′-dichlorodihydrofluorescein diacetate (H2DCF-DA), which forms the anion H2DCF^−^ and is oxidized to DCF (highly fluorescent). Treatments with antioxidant applications should decrease fluorescence in comparison to controls.

Nano- and microtechnology make it possible to encapsulate antioxidant compounds in polymeric or nonpolymeric matrices (nanoparticles, nanocapsules, and nano- and microemulsions, to name a few) [8]. This growing interest in nanotechnology includes electrospinning, which enables the relatively simple production of a continuous nonwoven nano- or microfiber of a controllable diameter using a capillary through which a polymer solution is pumped under a high-voltage source and a collector. The pendant drop of polymer solution, which is held by surface tension forces at the tip of the capillary, is electrified by the high voltage. As a result, electrostatic repulsion is established between similar charges within the polymer solution, resulting in Coulomb forces due to the external field. When the electrostatic repulsion force exceeds a threshold value that overcomes the surface tension of the polymer solution, a jet is produced from the pendant drop [9]. The jet undergoes stretching and whipping while traveling toward the collector. The solvent evaporates during this process, and then a solid nonwoven fibrous matrix is deposited on the collector [10]. According to Huang et al. [11], electrospinning seems to be the best process for nanofiber production in comparison to drawing (a process where only viscoelastic materials can be supported), self-assembly (which is more time consuming), template synthesis (which does not produce one-by-one continuous fibers), or phase separation (which, like self-assembly, takes more time than electrospinning does). Among other things, such as voltage, flow rate, polymer concentration, or needle–collector distance, one factor that affects fiber characteristics is the kind of polymer matrix, and considering the large number of synthetic polymers, semisynthetic polymers, and biopolymers available, an appropriate selection must be made by taking the environmental situation into account. Biopolymers such as bacterial polyhydroxybutyrate (PHB) or silk fibroin have been studied as materials for electrospinning production with promising results, getting round and smooth fibers with excellent morphology and biocompatibility for possible technological applications [12,13]. It has also been well reported that solvent and the addition of drugs or active compounds to nanofiber raw solution may change the diameter, viscosity, and conductivity of the fibers. In addition, this could also modify their chemical and physical capacity to adsorb and deliver drugs, as well as modify their mechanical properties [14,15,16].

The present study reviews articles reporting on the use of electrospun nanofibers loaded with antioxidant compounds. The purpose of this review is to identify both the major applications of antioxidant nanofibers and chemical compounds and the trends associated with these new technologies.

## 2. Reviewed Literature

The Scopus database was used to search and select articles related to three keywords: “antioxidant”, “nanofiber”, and “electrospun”. One-hundred and twelve publications belonging to various subjects areas were found, and 75% of the reviewed publications were papers written between 2016 and 2018, lending support to the innovative nature of this review. The excluded articles were related to the search, but they did not correspond exactly with the review subject. In fact, in some of them, the word “antioxidant” was found only in the abstract. Finally, 71 publications (both from the search and added by experts) from different journals from 2008 onwards were selected and conveniently categorized to conduct the review.

## 3. Electrospun Nanofibers: Development and Applications

Seven categories of antioxidant compound-loaded electrospun nanofibers in different areas such as food, chemistry, and (mainly) medicine were identified in this review, and they were defined as “wound dressings”, “tissue engineering”, “nanoencapsulation”, “food”, “stem cells”, “polymer-free”, and others.

Additionally, the eight articles reviewed in this first section (Table 1) did not declare any specifically defined application. In general, the medical, pharmaceutical, and food industries were commonly proposed as areas of application. In terms of Table 1, Section 3.1 and Section 3.2 respectively present a description of polymers used to obtain electrospun nanofibers and active compounds. Section 3.3, Section 3.4, Section 3.5, Section 3.6, Section 3.7, Section 3.8 and Section 3.9 present those articles with a defined application and are focused on the release and antioxidant activity of the nanofibers produced.

### 3.1. Polymeric Matrices of Electrospun Nanofibers

In the eight cited articles, different polymers were used regarding the purposes of fibers. Poly(lactic acid) (PLA) was selected by Chuysinuan et al. [24], and it is commonly used either individually or blended as a copolymer. Regarding its biodegradability and food compatibility, PLA is used for fibrous material preparation with biomedical purposes. Poly(lactic acid) copolymers degrade faster than pure polymers; therefore, copolymers such as poly(l-lactide-*co*-d,l-lactide) (coPLA) or poly(lactide-*co*-glycolide) (PLGA) are also used to obtain electrospun nanofibers, protecting loaded drugs from biological degradation, enhancing stability and sustained release, and allowing for a reduction in the administered dose [25]. Polyvinylpyrrolidone (PVP) and polyvinyl alcohol (PVA) are commonly used in drug delivery, have good biocompatibility, and are nontoxic and amorphous in nature. PVP is water-soluble, whereas PVA solubility depends on the temperature of the solution and the degree of polymerization [17]. Polyethylene glycol (PEG) is a polyether that is preferred due to its hydrophilicity. In the particular case reviewed, it was used for an expected hydrogen bond in the –O– group with the active compound [19]. Pluronic is a copolymer composed of poly(ethylene oxide)–poly(propylene oxide)–poly(ethylene oxide) (PEO–PPO–PEO): this structure gives pluronic a hydrophobic (PPO) and a hydrophilic section (PEO), which is why it is used in hydrophobic or water-insoluble drug delivery [20]. Gelatin can be classified as a semisynthetic polymer due to its origin in the hydrolysis of collagen (natural polymer). It is widely used due to its biocompatibility, biodegradability, availability, and low cost. Gelatin is water-soluble, and thus crosslinking it with different synthetic compounds contributes to its use in aqueous media; but, in turn, this can result in a loss of mechanical properties [21]. Polyacrylonitrile (PAN) is a polymer with interesting mechanical properties, stability, and carbon yield. Modified PAN is a technological precursor with various applications in medical, textile, and agricultural fields, among others. Electrospun fibers of PAN have small diameters and uniformity and are normally solved in polar solvents such as *N*,*N*-dimethylformamide (DMF) [26].

Pullulan is a natural, nontoxic polysaccharide produced by a fungus with flexible characteristics: it is a biopolymer (directly synthesized by living beings such as proteins or polysaccharides) and has commonly been tested for drug delivery, among other applications [20,27].

### 3.2. Antioxidants

Loaded active compounds were selected in the initial search (such as antioxidant compounds), and several types of antioxidant compounds were used in the cited articles as pure compounds or extracts.

#### 3.2.1. Tannic Acid (TA)

This polyphenol (from plants) appeared in three works due to its antimicrobial, antioxidant, and anticarcinogenic activities. It is water-soluble and interacts with various substrates, either through hydrogen bonds or by forming complexes such as TA/Fe^2+^. Fe^3+^ can attach to one, two, or three molecules of TA, and it has been tested for drug delivery and other applications. Loaded on PVA, the bi-(TA/Fe^3+^ (II)) complex shows the best prevention of bead formation, reinforcing nanofibers, while the complex formation does not affect tannic acid antioxidant activity [22]. Attached by hydrogen bonds to PEG, TA was studied by Zhou et al. [19], as they assembled thin nanofiber layers in a multilayer system and studied its behavior in terms of the release of TA (to exploit its benefits). The release of TA was found to be directly dependent on pH and temperature, thus increasing pH and/or temperature, resulting in increased release and TA maintaining its antioxidant capability. However, a comparative antioxidant activity related to pure TA was not reported. Tannic acid as a polyphenol was also studied along with three monophenols (ferulic (one hydroxyl group), caffeic (two hydroxyl groups), and gallic (three hydroxyl groups) acid) to crosslink gelatin. Total phenol content (TPC) in nanofibers and the antioxidant activity showed that gallic and ferulic acids seemed to react with gelatin noncovalently, and tannic acid presented a high TPC due to the quantity of hydroxyl groups. As expected, antioxidant activity increased while TPC increased, except in the case of tannic acid due to its hydroxyl groups [21].

#### 3.2.2. Rutin

This is a flavonoid formed from plant use as an antioxidant due to its 10 hydroxyl groups and phenolic groups, and it is also as a preventive agent of blood vessel diseases. It is not very water-soluble, which reduces its absorption in the digestive system, decreasing its bioavailability. Lee et al. [20] loaded rutin (RU) and pluronic (PL) on pullulan (PUL) nanofibers. The diameter of the RU/PL/PUL fibers was larger than fibers with two of those compounds. In terms of antioxidant activity, it was found that the presence of pluronic allowed for an acceptable rutin performance in a 2,2′-azino-*bis*(3-ethylbenzothiazoline-6-sulphonic acid) (ABTS) scavenging test compared to fibers without PL. The RU/PL/PUL nanocomposite showed similar scavenging compared to raw rutin.

#### 3.2.3. Poryphyrin

This is a large heterocyclic molecule belonging to tetrapyrroles. This antioxidant compound was loaded in two forms (5,10,15,20-tetrakis(*N*-methyl-4-pyridyl)-21,23*H*-porphyrin (TMPyP) and 5,10,15,20-tetrakisphenyl-21H,23H-porphyrin (TPP)) in a polyacrylonitrile (PAN) solution, and the ability to control the location of this compound on the electrospun nanofibers (produced due to its charge) was tested. The positional control of TMPyP was easier than TPP [23].

#### 3.2.4. Green Tea Extract

Depending on the processing technique, tea (*Camellia sinensis*) can develop into green tea. Green tea has various beneficial compounds, such as phenolic acids, theaflavin, caffeine, and theanine, while its major antioxidant activity is attributed to flavanols or catechins. To exploit the benefits of catechins, it is necessary to enhance their bioavailability and load green tea extract (GTE) on a PVP matrix for nanofiber production [17]. GTE reduces the fiber diameter in comparison to PVP; moreover, the DPPH assay showed that antioxidant activity increased with an increase in GTE concentration. GTE/PVP fibers did not achieve GTE activity, and the GTE/PVP film achieved an activity similar to the GTE concentration medium.

#### 3.2.5. Rice Extract

Three kinds of rice extract (riceberry, rice bran, and LehumPhum rice extracts) were loaded on PLA. Rice bran contains vitamin E and phenolic acids, which provide antioxidant and antitumor activities. With a DPPH assay, extracts of riceberry, rice bran, and LehumPhum rice resulted in 93%, 94%, and 97% antioxidant activity, respectively, while loading them on nanofibers resulted in the same order: 33%, 28–35%, and 38%. This was mainly due to rice extract release, which, with the three extracts, reached a maximum of 24% at 300 min and 33% at 2880 min. LehumPhum rice presented the highest antioxidant activity due to its anthocyanin content [24].

#### 3.2.6. Garcinia mangostana Extract

*Garcinia mangostana* is a fruit traditionally used for medicinal treatments such as ulcers, infected wound treatments, diarrhea, or abdominal pain in tropical regions of Africa and Asia. Its active compounds are xanthones (of which α-mangostin has the highest proportion), and due to its polyphenolic structure, it presents with high antioxidant properties. *Garcinia mangostana* extract (GME) was blended with PVP to produce nanofibers to enhance the bioavailability of GME [18]. The amount of GME was fixed, and PVP concentrations changed from 5–10%. A comparison between DPPH scavenging of GME extract and loaded GME/PVP fibers showed a slight rise in the antioxidant activity of GME/PVP fibers, and the thinnest one presented the greatest antioxidant capacity, which was probably related to its larger surface area. The GME release assay was unfortunately not published in the same article.

### 3.3. Wound Dressings

Wound dressings are an attractive option for wound healing. Attempts to develop a dressing that provides a proper healing environment have included fluid absorption, thermal isolation, and drug delivery, always considering care of the surrounding skin. The ability to protect wounds against microbial growth is very important in preventing bacterial proliferation [28]. Likewise, low values of the water contact angle and rapid absorption of the droplet by the dressing material are important properties in favoring the absorption of exudation in open wounds. According to Zahedi et al. [29], there are three kinds of wound dressings: (i) gauze or tulle, passive wound dressings that offer only coverage for wound rehabilitation; (ii) interactive material dressings, which offer permeability and gas transfer and act as a barrier against bacteria (hyaluronic acid (HA) or hydrogels are examples of this type); and (iii) bioactive material dressings, of which hydrocolloids and chitosan are good examples. As free radicals impede the normal and natural recovery of tissues, proper infected and noninfected wound care and healing can be enhanced by the action of antioxidant compounds that act as free radical scavenging agents [5,6,7], promoting tissue recovery and reducing healing times. Sixteen articles were found and are reviewed in this section, as shown in Table 2.

Charernsriwilaiwat et al. [30] produced and studied electrospun *Garcinia mangostana* extract (GME) loaded on chitosan–ethylenediaminetetraacetic acid/polyvinyl alcohol (CS–EDTA/PVA) nanofibers. Chitosan is a biopolymer that is present in the exoskeleton of some crustaceans, such as crabs and lobsters. GME showed a rapid release (80% in 60 min) due to matrix erosion in the medium and swelling behavior of the matrix. This could be a disadvantage considering that controlled and time-sustained releases are expected from a wound dressing. Antioxidant and antibacterial activities were retained, accelerating the healing process. Silk fibroin (SF) is a protein-based, biodegradable, and biocompatible material, the mechanical properties and electrospinnability of which can be enhanced by blending it with other polymers. Silk fibroin is produced by spiders, mites, scorpions, bees, and silkworms. It is spun into fibers in the metamorphosis phase [45]. The domestic silkworm *Bombix mori* is the most common source of silk production. Three articles in this section used silk fibroin. Lin et al. [12] produced nanofibers of grape seed extract (GSE)-loaded SF/PEO. GSE contains a mixture of polyphenols that is mainly catechin, epicatechin, epicatechin gallic acid ester, and gallic acid. GSE is used in medicine and the cosmetic industry due to its antioxidant capability. The GSE release in the named work was divided into three phases and reached 65% after 350 h. Antioxidant activity improved with the increase in GSE concentration, except for the medium level (3%), and showed the best results with significant differences compared to the negative control.

Another natural plant used as an antioxidant, fenugreek, was incorporated as a seed extract into silk fibroin nanofibers by Selvaraj and Fathima [31] for wound healing applications. A DPPH assay revealed that at 30 min of incubation, silk fibroin fibers showed 5.6% scavenging capacity, while fenugreek/silk fibroin fibers reached 49.3%, confirming the effect of fenugreek seeds. Furthermore, 58% DPPH scavenging in the first hour and 67% at the end of 24 h confirmed the incubation time dependence. This was clearly related to the release profile, because at 24 h, fibers with a higher fenugreek concentration presented a cumulative release of 73% ± 0.9%, while fibers with a lower fenugreek concentration reached only a 21.5% ± 0.9% release. The article presented by Basal et al. [32] reported on olive leaf extract-containing polyphenols loaded on silk fibroin/hyaluronic acid nanofibers. The biopolymer hyaluronic acid is a natural polysaccharide present in skin and connective tissues, among other things. Trolox, with an equivalent antioxidant capacity, was used to determine antioxidant activity, with similarities found between total phenol content and olive leaf extract. Therefore, this capacity was attributed to rapid-release polyphenols in the extract, since the olive leaf extract was weakly attached to fibers. Polycaprolactone (PCL) is an aliphatic polyester that is also biodegradable and biocompatible, whose elasticity means it may be considered a proper material for wound dressings. However, its hydrophobicity needs to be taken into account for use in this area. Ajmal et al. [33] developed nanofibers of PCL containing ciprofloxacin HCl (CHL), a fluoroquinolone antibiotic used for wound dressings [46] and quercetin. Quercetin (Que) is a flavonoid found in plants, fruits, seeds, and leaves. It is not very water-soluble, and its bioavailability is therefore reduced; nevertheless, it possesses recognized antioxidant activity. This property was tested in vitro using a DPPH radical inhibition. Regarding the presence of hydroxyl groups in both PCL and ciprofloxacin HCl, PCL/CHL showed near 13% radical scavenging, while Que incorporation into fibers resulted in 40% radical inhibition due to a greater amount of polyphenols in quercetin, which allowed PCL/CHL/Que fibers to protect the red blood cell membrane against lipid peroxidation, helping to maintain its functionality. PCL was also employed by Ravichandran et al. [34], who loaded it with *Clerodendrum phlomidis* leaf extract. *C. phlomidis* is a well-known bush that is mainly present in Africa and India, and it is active against oxidation. A DPPH assay was carried out in order to determine the antioxidant activity of the resulting mats, and the results were equivalent to the *C. phlomidis* extract (both between 68.4 and 69.4 μg/mL), confirming the stability given by the electrospinning process. Approximately 35% of the extract from nanofibers was released in 24 h. Rather et al. [35] also used PCL as a matrix for wound healing: they blended it with gelatin and loaded it with cerium oxide, which possesses antibacterial and antioxidant activities. Nanoparticles of cerium oxide (CeNPs) contained in electrospun nanofibers of PCL/GE were tested as reducers of ROS levels produced by oxidative stress. A 2′,7′-dichlorodihydrofluorescein diacetate (DCFDA) assay was carried out: PCL/GE nanofibers decreased by 12% the fluorescence intensity of DCF, while PCL/GE/CeNPs reached 30% diminished intensity, increasing cell viability compared to the control due to a reduction in oxidative damage. Deldar et al. [36] loaded a natural flavone called chrysin on a PCL/PEG matrix, with no significant differences in terms of release during the first 8 h for two different concentrations. After that time, nonrelative quantities of released chrysin were reported. Antioxidant activity related to the ability of chrysin to inhibit free radical damage was studied using fluorescence signals, where both chrysin concentrations (5% and 15%) showed excellent results. It is worth noting that the fiber with the highest amount of chrysin maintained its antioxidant activity longer. Grape seed extract (GSE) was also used by Locilento et al. [37]: they used PLA and PEO as a polymeric matrix and prepared electrospun nanofibers. The release profile showed an abrupt release by 60% in 24 h from PLA, while a more controlled release from PLA/PEO reached 20% in 24 h, maintaining the release for a more prolonged time. Antioxidant activity was tested with a DPPH radical scavenging assay, showing near 85% radical inhibition even after 45 days. Curcumin (Cur), from the medicinal plant *Curcuma longa*, is recognized for its therapeutic properties, such as its anti-inflammatory, antioxidant, and antibacterial activities. Cur-loaded coPLA/PEG fibers (in a study by Yakub et al. [38]) resulted in the diameter of the fibers decreasing with the curcumin concentration in an inverse proportion. In this case, microfibers turned out to be morphologically defect-free. PEG was added to increase hydrophilicity: it also increased the Cur release due to its hydrophilic nature, but this also resulted in a negative effect on the antibacterial activity of Cur, which was found to be lower than in coPLA/Cur fibers. The antioxidant activity was evaluated spectrophotometrically and was reported as a DPPH radical inhibition in solution. Cur reached 71%–72% scavenging capacity, turning radicals into a stabilized compound. Pankongadisak et al. [39] also studied curcumin and developed polylactic acid fibers containing Cur fibers. While Cur concentration in the mats and the immersion time in Phosphate-Buffered Saline (PBS) increased, Cur release increased, reaching nearly 58%. Antioxidant activity tested by a DPPH assay resulted in 42–53%; in addition, curcumin showed diminished radical inhibition capacity, while the immersion time increased due to curcumin instability. Peršin et al. [40] also studied olive leaf extract. They loaded the extract on a carboxymethylcellulose/sodium alginate/CaCl_2_/polyethylene oxide polymeric solution. Carboxymethylcellulose is a cellulose derivative and sodium alginate comes from brown seaweeds (both are widely used in the food industry). Using the DPPH assay of the extract, the maximum obtainable radical inhibition was determined to be around 90%. Mats without the extract did not present antioxidant activity; nonetheless, when the extract was incorporated into the mats, 81% radical scavenging capacity was found. This means that 89% of olive leaf extract is usable when it is loaded onto electrospun nanofiber mats. Wutticharoenmongkol et al. [41] made use of cellulose acetate (a common polymer fabricated from cellulose) to load gallic acid (GA) for antioxidant electrospun nanofiber production. Gallic acid release in normal saline buffer resulted higher for 20% GA content than 40% GA content, and the maximum reached 96% at 2880 min; on the other hand, transdermal diffusion through pig skin was assessed, with a maximum of 87% release in normal saline buffer. Antioxidant activity reached 78–85% of DPPH radical scavenging. Poly(vinyl alcohol) nanofibers loaded with honey in three different concentrations were reported by Sarkar et al. [42], with no significant changes in the electrospinning parameters. In the DPPH assay, fibers with 0.5% honey showed the highest antioxidant activity. The capability of honey to inhibit DPPH radicals was proportionally maintained after the electrospinning process, despite slightly decreasing, and it presented with epithelial cell adhesion and proliferation. Tang et al. [43] also loaded honey onto nanofibers: they used an alginate/PVA combination as a matrix, achieving approximately 66% DPPH radical inhibition after 9 h for nanofibers containing 20% honey, and antioxidant activity increased over time in terms of honey release. Considering its antibacterial activity and biocompatibility, this material was proposed as part of a wound dressing. Amna et al. [44] encapsulated capsaicin (from chili pepper, which is often used as an analgesic) in polyurethane (PU) (which is commonly used as foam for pillows, mattress, and thermal coatings, among other things): 50% capsaicin was cumulatively released from fibers in 18 h after 15% were burst-released at the beginning. A DPPH scavenging of 70% was obtained from capsaicin; however, capsaicin loaded on polyurethane reached 78% DPPH radical inhibition, confirming enhanced stability and antioxidant capacity. They also tested the intracellular ROS (radical scavenging species) in muscle cells using fluorescence microscopy and found that the amount of apoptotic cells was reduced by treatment with capsaicin.

### 3.4. Tissue Engineering

In tissue engineering (TE), it is expected that tissue formation (by culturing cells in situ on a scaffold miming an extracellular matrix) assists in cell attachment and proliferation. The main problems in TE are the body’s rejection of inserted scaffolds and the excessive formation of reactive oxygen species (ROS) causing oxidative damage. The antioxidant activity in TE is significant in terms of reducing ROS to prevent or reduce this oxidative damage, which impedes cell growth. Several bioactive compounds with antioxidant activity plus anti-inflammatory and/or antibacterial properties, biodegradability, and biocompatibility were tested in the research articles shown in Table 3 to accelerate or enhance the time needed for new tissue formation. Additionally, for skeletal muscle tissue engineering (SMTE) and large-scale applications, the scaffold material should be moldable to various desired formats [47].

Manchineella [47] and Nune et al. [48] studied melanin (a natural polymeric pigment) blended with silk fibroin (SF) to obtain nanofibers with conducting properties, antioxidant activity, biodegradability, and biocompatibility. Silk and melanin/silk nanofibers were produced, and oxidative stress control was assessed by measuring the reactive oxygen species (ROS) in myoblasts cultured on scaffolds in one work, while in the other, antioxidant activity was determined with a DPPH assay for random and aligned fibers. Qualitatively, melanin reduced the intracellular ROS levels of the myoblasts, improving cell proliferation in contrast to SF nanofibers; also, fibers containing melanin showed DPPH radical scavenging above 40% in the case of randomly aligned fibers. The release profile was not studied in any case. Kandhasamy et al. [49] also used silk fibroin for skin tissue engineering: they loaded this biopolymer with a synthetized quinone-based chromenopyrazole (QCP). Antioxidant activity was determined by DPPH assay, which measured the absorbance of the resulting solutions. The half maximal inhibitory concentration (IC_50_) of the DPPH scavenging obtained was near 5.5 μg, and the fibers allowed for strong cell growth. The antioxidant activity of spirulina, a prokaryotic microalga, is linked to the phycocyanin compound, among other things. This property adds to its anti-inflammatory and anticancer activities. Its biocompatibility led to spirulina loaded on nanofibers being studied for tissue engineering purposes. Spirulina was tested on a PCL-based scaffold to provide the necessary environment for cell proliferation and subsequent tissue regeneration. Spirulina showed interesting results, reducing oxidative stress, assisting fibroblast cultures and favoring tissue formation [50]. PCL was also loaded with spirulina by Kepekçi et al. [51], and this improved the hydrophilicity of fibers in line with its concentration. The uniformity of fibers was found to be better, with a lower concentration of the active compound. Increased antioxidant activity resulted when the spirulina concentration increased. At 3% (mass/volume) concentration, the antioxidant activity was similar to 6% (mass/volume). This could have been related to a smaller diameter of fibers using 3% (mass/volume) of spirulina, providing a larger contact area. The release profile of phycocyanin also showed similarities between 3% and 6% spirulina concentrations, which could have been for the same reason.

Quercetin (Que) and curcumin (Cur) were individually loaded onto PCL and blended with poly(hydroxybutyrate-*co*-hydroxyvalerate) (PHBV) by Da Silva et al. [52]: they intended to use the scavenging capacity of free radicals and chelate metal ions to develop a scaffold to stimulate tissue recovery. PHBV is a copolymer of bacterial origin belonging to the polyhydroxyalkanoate (PHA) group. Several bacteria can synthetize PHB, PHV, and PHBV from different carbon sources under nutrient limitations. The copolymer can enhance the mechanical properties of each polymer on its own [55,56,57], while the polymer structure provides protection against the degradation of active compounds. Que/PCL, Cur/PCL, Que/PCL/PHBV, and Cur/PCL/PHBV nanofibers were obtained from the electrospinning process. DPPH and ABTS assays were used to assess the antioxidant activity of nanofiber mats produced with fixed quantities of Que and Cur. Curcumin-loaded PCL and PCL/PHBV presented a more rapid inhibition of the DPPH radical, but a sustained assessment showed that Que/PCL reached almost 87% inhibition at 40 min, while Cur/PCL reached 44% inhibition on average. With the ABTS assay, rapid scavenging for 6, 12, and 18 min did not show any significant difference for any formula. Quercetin both in PCL and PCL/PHBV reached close to 64% ABTS reduction, while Cur/PCL and Cur/PCL/PHBV reached nearly 32% and 65% on average, respectively.

Phosvitin (PV) is a protein with metal-chelating properties that bonds with iron, inhibiting Fe^2+^-catalyzed hydroxyl radicals production. In addition, tannic acid (TA) can interact with several materials given its structure and functional groups and is considered to be a good ligand for metal ion coordination. These two compounds have free radical scavenging capacity and were blended with cellulose [53] (the most abundant biopolymer on earth): nanofiber scaffolds for tissue engineering were obtained. The antioxidant activity of nanofiber mats was evaluated with regard to their scavenging effect against superoxide, DPPH, and hydroxyl radicals. Cellulose did not show any scavenging activity against superoxide or hydroxyl radicals, while against DPPH it showed less than 10%. However, TA/PV loaded on cellulose presented almost 90% scavenging capacity against DPPH for every concentration of TA/PV; between 23% and 35% scavenging against superoxide radicals; and between 56% and 70% scavenging against hydroxyl radicals. The amount of loaded TA/PV determined the amount of scavenging in superoxide and hydroxyl radicals. However, layer-by-layer mats were obtained, and as TA was the outermost component, superoxide scavenging was lower than expected and hydroxyl scavenging was higher than the trend. This could have been because of superoxide radical scavenging being higher for PV than for TA and because of hydroxyl radical scavenging for TA being higher than for PV.

Llorens et al. [54] studied vitamin B_6_ in the form of pyridoxine, pyridoxal, *p*-coumaric acid, and caffeic acid as excellent antioxidant compounds able to inhibit oxidative DNA damage loaded on PLA nanofibers. The same active compounds were again blended with PLA to assess the prevention of oxidative stress in cells when proliferation was carried out on scaffolds. The effect of nanofibers with antioxidant properties was assessed against 2,2′-azobis(2-methylpropionamidine) dihydrochloride (AAPH) as the agent of oxidative damage. AAPH doses between 1.9 and 5.9 mM decreased cell viability almost 30–60%. Unloaded fibers did not show any significant difference from these results. The higher dose did not show significant differences from the control either. Nevertheless, nanofibers containing pyridoxine, pyridoxal, *p*-coumaric acid, and caffeic acid significantly inhibited the AAPH oxidative damage, with no differences between them.

### 3.5. Nanoencapsulation

Encapsulation is an alternative that enhances the bioavailability and solubility of some active compounds. In addition, this technique provides for a controlled release and protection against external agents that could degrade compounds or decrease their activity. Technological advances have allowed scientists to focus on nanoencapsulation. Nanomaterials (nanofibers included) are materials on a nanometric scale that have been studied and developed because of their high specific surface area and mechanical properties, and they offer diverse applications such as drug delivery (antibiotics, antibacterial agents, anticancer and anti-inflammatory drugs), implants, or food packaging. The papers that specifically defined nanoencapsulation and its possible application areas in terms of the reviewed composites are presented in Table 4.

In a work by Aytac et al. [58], a quercetin/gamma-cyclodextrin inclusion complex was loaded on zein to obtain nanofibers. Zein is a protein storage of maize that is resistant to microbial attack. When an inclusion complex between Que and cyclodextrin is created, the availability of Que increases; thus, its properties are better used. With a DPPH assay, the antioxidant activities of zein, Que/zein, and complex/zein nanofibers were determined. When the Que concentration was increased, the antioxidant activity increased, too, and the complex formation enhanced the ability of Que to inhibit DPPH radicals. On the other hand, zein did not present with antioxidant activity. Sour cherry has a high polyphenol content (antioxidant and sensitive compounds). Encapsulation allows for the protection of polyphenols; thus, sour cherry extract was both uniaxially and coaxially electrospun with gelatin and gelatin–lactalbumin by Isik et al. [59]. Total phenolic, flavonoid, and anthocyanin contents were determined and showed a direct relation with antioxidant activity. Uniaxial electrospun fibers presented the highest antioxidant activity, with no significant differences from the electrospun samples. Vitamin E is mainly composed of α-tocopherol (α-TC) and presents with recognized high antioxidant activity, and it is used in wound dressings and drug delivery. Aytac and Uyar [60] prepared an α-tocopherol/cyclodextrin inclusion complex (α-TC/CD) for its inclusion in PCL nanofibers: α-TC/CD/PCL nanofibers and α-TC/PCL nanofibers showed similar antioxidant activity in methanol. In this case, the DPPH assay was carried out in methanol/water (1:1), where the nanofiber-encapsulated complex showed a significantly higher inhibition of DPPH radicals. The natural phenolic monoterpene thymol possesses antioxidant and antimicrobial properties [67]. However, it is a volatile compound and is almost water-insoluble. To enhance its availability and preserve its antioxidant properties, Amariei et al. [61] developed a thymol nanoencapsulation process with poly(acrylic acid)/poly(vinyl alcohol) (PAA/PVA) in electrospun fibers, which were later grafted with poly(amideamine) and subsequently with thymol. Poly(amideamine) was attached to PAA/PVA by bonding its –NH_2_ group with carboxyl groups, forming amide. Thymol release turned out to be pH-dependent and presented a controlled delivery. Since inclusion complexes contribute to the stability of active compounds, Aytac et al. [62] studied the complexation of gallic acid (GA) with cyclodextrin (CD). Given its antioxidant and antimicrobial properties, GA was selected to form a complex with CD and was loaded onto PLA in nanofiber conformations. Once again, a DPPH assay was used to determine the antioxidant activities of PLA nanofibers, GA/PLA nanofibers, and complex/PLA nanofibers. The average DPPH radical inhibition was 4%, 95%, and 96%, respectively. The similar results between complexed and noncomplexed gallic acid might have been related to GA solubility and the position of GA in CD cavities. Ferulic acid loaded onto PLGA/PEO samples was electrospun to obtain nanofibers to encapsulate the active compound. It was observed that ferulic acid settled in the core of the nanofibers. The capability of scavenging DPPH radicals was 41% and 59% of DPPH inhibition, respectively, for isolated ferulic acid compared to encapsulated ferulic acid [63]. A work by de Freitas Zômpero [64] presented β-carotene encapsulated in nanoliposomes and then loaded onto PVA and PO electrospun fibers to enhance the bioavailability of β-carotene. The protection ability of the photosensitive antioxidant molecule was tested upon exposure to ultraviolet light. When β-carotene was encapsulated in the nanoliposome and polymer fiber, its photo-oxidation decreased, enhancing its stability over time. Nearly 87% of the initial β-carotene remained in the PEO nanofibers and 80% in the PVA nanofibers at 6 h of light exposure. The antioxidant activity was maintained as well. The nanoencapsulation of the incoming polyphenols into poly(2-hydroxyethyl methacrylate) (pHEMA) was reported by Ghitescu et al. [65] with good results. Gallic acid (GA), syringic acid (SA), vanillic acid (VA), catechin (Ca), and spruce bark extract were selected to load pHEMA. The scavenging of DPPH radicals was used to assess the antioxidant activity of nanofibers. The most active polyphenols were those with three hydroxyl groups, and the least active ones, spruce bark extract and vanillin, were tested for one week, free and encapsulated, to determine the effect of encapsulation on time-sustained antioxidant activity. Vanillic acid was highly unstable compared to the spruce bark extract. Encapsulation provides protection and extends the lifespan of antioxidant compounds. Finally, Hussain et al. [66] reviewed trends related to curcumin nanoencapsulation, and according to their results, when a hydrogel nanocomposite was developed based on methoxypoly(ethylene glycol) (MPEG)/PCL, alginate, chitosan, and curcumin, antioxidant activity increased only slightly. For nanohybrid scaffolds of collagen, alginate, chitosan, and curcumin, there were increased anti-inflammatory and antioxidant activities, and the nanoformulation of MPEG/PCL, linoleic acid, tween-20, chitosan, and curcumin slightly improved the antioxidant capability of curcumin.

### 3.6. Food

Functional foods, food packaging, and food preservation are proposed as applications in this area. Biocompatibility, antioxidant, and antibacterial properties are necessary for food preservation. The main cause of food deterioration is related to the oxidation of carbohydrates and lipids; therefore, antioxidants can provide a solution to this issue. Different antioxidant compounds that have been loaded onto a polymeric matrix are shown in Table 5.

Gliadin is a glycoprotein present in some cereals such as rye or wheat, with low solubility in water. It was used by Akman et al. [68] for nanofiber development containing curcumin in order to develop a material for the food industry due to the antimicrobial and antioxidant activities of this active compound. Curcumin was dispersed thoroughly into nanofibers and was released in two stages: a rapid release of 25% curcumin in the first hours was followed by a controlled release until 95–99% at 48 h. Besides, the most concentrated fibers were released above 30% in the first 3 h. Antioxidant activity tested by DPPH assay showed between 56% and 91% DPPH radical scavenging related to curcumin concentration. By increasing the curcumin concentration in fibers, radical inhibition also increased. Sharif et al. [69] also used gliadin to load inclusion complexes of ferulic acid (FA) with hydroxypropyl-beta-cyclodextrin (HP-β-CD-IC). With various FA concentrations, the DPPH scavenging of fibers was assessed. The incorporation of FA into fibers did not affect its antioxidant capacity, which resulted in 92% DPPH inhibition (gliadin/FA resulted in nearly 91%, and gliadin/FA/HP-β-CD-IC was nearly 89%). This material was also proposed for biomedical applications. The biopolymer PHBV was used as matrix for oregano essential oil (OEO) (with polyphenols in its conformation due to its antioxidant and antimicrobial activities), rosemary extract (RE) (obtained from the aromatic plant with antimicrobial properties), and green tea extract (GTE) separately. The resulting nanofibers electrospun by Figueroa-Lopez et al. [70] were proposed as biopackaging to prolong the life of foods. OEO resulted in the highest DPPH inhibition, reaching nearly 92%, which was related to its polyphenol content; RE reached nearly 74% DPPH scavenging; and GTE resulted in approximately 72%. When essential oils and green tea extract were electrospun with PHBV, DPPH radical scavenging decreased, reaching approximately 43%, 26%, and 22%, respectively, from OEO, RE, and GTE. Over time, DPPH inhibition also decreased: when it was measured at 15 days, the results were approximately 9.5%, 7%, and 6% for OEO, RE, and GTE, respectively, which could have been related to a release of volatile compounds. Active compound releases were not assessed. Fonseca et al. [71] made use of soluble potato starch as a matrix for carvacrol in order to develop fibers for food packaging. Potato starch is a biopolymer that is commonly available and inexpensive. Carvacol is a phenol present in oregano and pepperwort, among other plants, and it has recognized antimicrobial activity [67] and antioxidant properties. ABTS radical inhibition was used to measure the antioxidant activity of carvacol, which resulted in the highest radical inhibition reaching 93%. When carvacol was loaded into fibers of potato starch at 30% and 40% (v/v), it achieved 76% and 83% ABTS radical inhibition, respectively. Chitosan/PVA with fish-purified antioxidant peptide (FPAP) was electrospun by Hosseini et al. [72]: the antioxidant activity of FPAP was attributed to its hydrophobic amino acids. This capacity of FPAP was determined for different FPAP concentrations using a DDPH assay, and the results were between 33% and 65%. When FPAP was loaded onto a chitosan/PVA matrix and electrospun, DPPH radical scavenging was between 15% and 45%, which could have been related to interactions between the polymers and the active compound, aside from the electrospinning process. It was concluded that increases in the FPAP concentration increase radical inhibition. Munteanu et al. [73] made use of PLA and chitosan along with argan and clove oils to obtain electrospun nanofibers. Argan oil (AO) is extracted from argan seeds, and clove oil (CO) is commonly known due to its anesthetic attributes. After 24 h, DPPH inhibition resulted in nearly 29% for CO and 17% for AO, and CO antioxidant activity was maintained even after 7 days of storage, confirming the encapsulation of CO in chitosan. Regarding the antibacterial properties of silver nanoparticles (Ag-NPs), Zhan et al. [74] mixed them with tannin acid/chitosan/tripolyphosphate (TA/Ch/TPP) and added them to a PVA/polyacrylic acid (PAA) matrix to obtain nanofibers. Tannic acid was used as a reducer compound, and tripolyphosphate was used to debilitate the possible hydrogen bonds between tannic acid and PVA. As free radicals are harmful to foods, a DPPH assay determined the antioxidant activity of the nanofibers produced. The scavenging rate of Ag/TA/Ch/TPP/PVA/PAA enhanced antioxidant activity in comparison to PVA/PAA nanofibers. Moreover, an increase in the TA concentration resulted in an increase in antioxidant activity. Yet, when Ag-NPs were included, the radical scavenging ability decreased, possibly due to a decrease in the TA proportion. Non-ionic Tween 80, cationic cetyltrimethylammonium bromide (CTAB), and anionic sodium dodecyl sulfonate (SDS) surfactants were added to a curcumin/gelatin system to enhance curcumin release, maintaining its antioxidant properties. Through DPPH and ferric-reducing antioxidant power (FRAP) assays, at 30 min, the antioxidant activity of curcumin/gelatin with and without surfactants exhibited more DPPH radical scavenging than did FRAP. In addition, CTAB and curcumin/gelatin radical scavenging activities did not differ, but Tween 80 and SDS presented significantly lower activity. Additionally, CTAB showed the highest FRAP (significantly higher than curcumin/gelatin), as CTAB favors curcumin release [75]. Gelatin, peppermint essential oil (PO), and chamomile essential oil (CO) were used by Tang et al. [76] for edible packaging material development. A DPPH assay was carried out to evaluate antioxidant activity, demonstrating that the antioxidant capacity of CO and PO was maintained after an electrospinning process. Considering a concentration of 9% for both essentials oils, the antioxidant activity of gelatin/CO (nearly 70% scavenging) was higher than that of gelatin/PO (near 50%), mainly due to the greater amount of polyphenols in chamomile essential oil. *Urtica dioica* L. is a medicinal plant traditionally used for its antiviral and antimicrobial properties. It is chemically composed of amino acids, omega 3, vitamin C, and β-carotene, among other constituents. *Urtica dioica* L. extract/PCL nanofibers were included in whey protein isolate (WPI) to preserve rainbow trout fillets during storage. Considering that lipid oxidation takes place in fish and that *Urtica dioica* L. possesses phenolic compounds with antioxidant activity, *Urtica dioica* L. extract loaded onto PCL nanofibers contributed to fish preservation during storage, significantly delaying lipid oxidation [77].

PCL was also found as a matrix for sage extract (SE) in the work of Salević et al. [78]. Sage is a bush commonly used in cookery, and it has recognized antibacterial and antioxidant activities. This last property was determined in fibers in terms of the inhibition of DPPH radicals, with a correlation that was found between the time of exposition and antioxidant activity being related to the release of active compounds. After 24 h, the antioxidant activity reached between 30% and 85% for PCL with SE in different concentrations (from 5–20%). In addition, γ-cyclodextrin (CD) is one of the three most used cyclodextrins: it is soluble and has good bioavailability with no side effects. Vitamin E (α-TC)/CD complexes enhance polymer solubility, reduce the α-TC diffusion rate, and protect food against oxidation, extending their lifetime. This complex blended with PLA was used by Aytac et al. [79] to produce electrospun nanofibers for food packaging materials. Through the DDPH method, the inhibition of DPPH radicals was determined for PLA, α-TC/PLA, and α-TC/CD/PLA nanofibers. Radical inhibition was 4%, 97%, and 97%, respectively, indicating that the antioxidant activity of loaded fibers depends on the presence of active compounds with high scavenging capacity. Vitamin E together with silver nanoparticles (Ag-NPs) (due to their antibacterial properties) was loaded onto PLA by Munteanu et al. [80] to obtain a multifunctional biomaterial. Vitamin E/Ag-NP/PLA fibers resulted in a smaller diameter than did PLA fibers. Antimicrobial activity was tested against *Escherichia coli*, *Salmonella typhymurium*, and *Listeria monocytogenes*, and the behavior of this nanocomposite yielded 100% inhibition at 48 h for each microorganism. Antioxidant activity showed 94% DPPH radical inhibition against 4% unloaded PLA. Given that Ag-NP release is very low, this material could be used for food applications. The tea phenolic compounds presented antioxidant activity and inhibition against microorganisms. Blends of tea phenolic compounds and PLA in different concentrations were electrospun by Liu et al. [81] to obtain nanofibers for food packaging. PLA did not show any significant antioxidant activity against DPPH radicals (almost 5% radical scavenging), and tea phenols/PLA in a 1:3 proportion showed the highest percentage of inhibition against DPPH radicals (95% radical scavenging on average). When the quantity of tea phenols increased, the morphology of the nanofibers changed; thus, a proper release could not be assured, and the antioxidant activity might be affected in this case. Finally, electrospun fibers of rosemary extract-loaded PVA [82] was proposed by Estevez-Areco et al. as packaging for hydrophilic and acid food products. PVA retained nearly 88% of rosemary polyphenols, and their release was found to be slower in aqueous medium due to their lipophilic characteristics. Antioxidant capacity was determined by DPPH scavenging, and with 10% rosemary extract, fibers reached 57% of DPPH radical scavenging.

### 3.7. Stem Cells

Since stem cells need a proper environment to reproduce and proliferate and since oxidative stress can generate cell damage, antioxidant compounds have been tested to assess their effects on the stem cell environment. Watercress extract (WE)-loaded PCL/PEG matrix nanofibers were developed by Dadashpour [83] to evaluate the effects on proliferation, adhesion, stemness preservation, and cytoprotection of adipose-derived stem cells (ADSCs). Watercress has potent antioxidants and vitamins, among other compounds: it is well known for its anti-inflammatory and antioxidant activities and is a proper environment for stem cell proliferation due to this antioxidant capacity. Radical scavenging capacity was determined through a DPPH assay. The antioxidant activity was time-sustained. Measured at different times, WE improved the inhibition of DPPH radicals, resulting in higher WE/PCL/PEG nanofibers compared to PCL/PEG nanofibers. Deldar et al. [84] conducted a similar study, developing nanofibers using the same PCL/PEG polymers loaded with chrysin (Chr). Chr/PCL/PEG nanofibers presented time-dependent, increasing antioxidant activity. Moreover, antioxidant activity was significantly higher for the Chr/PCL/PEG nanofibers than for the PCL/PEG nanofibers, confirming the input of the antioxidant activity of chrysin. When stem cells were exposed to H_2_O_2_ and cell viability was measured, the cytoprotective effect of Chr/PCL/PEG nanofibers on cells was confirmed when compared to PCL/PEG nanofibers. The effect of antioxidant compounds on preventing oxidative damage and enhancing cell proliferation is well recognized and has been demonstrated. In the papers in this section, stem cell proliferation proved the importance of free radical inhibition when stem cells are used.

### 3.8. Polymer-Free Electrospun Fibers (Special Section)

Electrospun nanofibers have traditionally been polymer-based. Polymer-free technology increases the active compound concentration, allowing for significantly smaller quantities of nanofibers to produce similar effects. Although this is not a nanofiber application itself, it has been included as a special section. Three articles with these characteristics were reviewed. Celebioglu et al. [85] reported on the development of an inclusion complex between vanillin and cyclodextrin in three forms (hydroxypropyl-β-cyclodextrin (HPβCD), hydroxypropyl-γ-cyclodextrin (HPγCD), and methyl-β-cyclodextrin (MβCD)) to find the best complex in terms of vanillin volatilization. The bioavailability of vanillin improves when it is included in the complex. Despite vanillin being widely known for its flavor, it has antioxidant properties related to its phenolic structure. A DPPH assay was performed to assess the radical scavenging activity. The poor antioxidant activity of pure vanillin was attributed to its high volatility. Vanillin preserved in the inclusion complexes improved the radical scavenging activity due to an increase in its solubility, which made vanillin available in the scavenging process. This composite has been proposed for food-related applications. Similarly, Aytac et al. [86] developed an inclusion complex from cyclodextrin (HPβCD, HPγCD, and MβCD) and geraniol as a flavor and fragrance delivery composite. Geraniol is also volatile, and polymer-based nanofibers do not preserve as much geraniol compared to its nonpolymeric-based composite. Geraniol exhibits antimicrobial, anti-inflammatory, insecticidal, antioxidant, and antifungal properties. Improving its solubility by means of an inclusion complex makes it possible to exploit its benefits. The DPPH analysis had similar results for vanillin. When radical scavenging activity was measured for the complexes, it was higher than geraniol radical scavenging activity due to its availability. Geraniol inclusion in nanofibers improved antioxidant activity in comparison to CD nanofibers. An upgrade of vanillin and geraniol was observed when they took part in the inclusion complexes: stability and availability was enhanced, and a more controlled release was enabled. Celebioglu et al. [87] reported the encapsulation in an inclusion complex of thymol and cyclodextrin in three forms (HPβCD, HPγCD, and MβCD) to enhance stability due to the volatile character of thymol. The inhibition of DPPH radicals showed a dependence on thymol concentration. The more concentrated the thymol was, the more radical scavenging there was. In addition, all of the complexes reached an average of 94% radical inhibition in contrast to the 25% exhibited by thymol under the same conditions. This was clearly attributable to polymer-free encapsulation.

Small molecules such as phospholipids have recently been used as matrices for electrospinning, with applications in cosmetics, drug delivery, and tissue engineering due to their specific permeability and biocompatibility, among other characteristics. Some authors, such as Jørgensen et al. [88] and Mendes et al. [89], have studied the effect of solvents and coaxial processing over phospholipidic fiber characteristics and have reported on the nanomechanical properties of asolectin electrospun fibers. Shekarforoush et al. [90] used the phospholipid asolectin, which presents antioxidant properties itself, as a carrier for vanillin and curcumin. Cumulative release in PBS at 37 °C was determined by UV-VIS spectrophotometry, and it reached 95% and 70% for vanillin and curcumin, respectively. Total antioxidant capacity was assessed with and without polyphenols (measured using the phosphomolybdenum method (the reduction of Mo (VI) to Mo (V))). Asolectin exhibited an antioxidant capacity between 71 and 89 μg of gallic acid equivalents/mg of fiber for different formulations. Antioxidant stability could be maintained for vanillin and curcumin encapsulated on asolectin in comparison to vanillin and curcumin by themselves.

### 3.9. Others

The following nine articles (presented in Table 6) do not belong to any other group. Likewise, two of these five were included in this review even though their applications barely fit the focus proposed to demonstrate the spectrum of polymeric electrospun nanofibers and antioxidant properties.

Vatankhah et al. [91] sought to develop a transdermal patch material using cellulose acetate and rosmarinic acid. Rosmarinic acid release in acetate buffer and antioxidant activity increased as its concentration in fibers increased. In vitro release reached 95% after 64 h for fibers containing 10% of the active compound, which meant an IC_50_ of 95 μg/mL. Rosmarinic acid maintained its antioxidant activity after the electrospinning process. Khoshnevisan et al. [92] presented a review of various articles related to cellulose acetate-based nanofibers. Regarding the antioxidant compounds used, curcumin, asiaticoside, (6)-gingerol, ferulic acid, alkannins, and shikonins were selected. These loaded compounds had no effect on morphology or the mean diameter of the nanofibers produced. The release profiles obtained made it possible to propose these types of composites as possible wound dressing materials. An implantable anticancer drug was proposed by Vashisth et al. [93] using quercetin loaded onto PLGA/PCL to increase its bioavailability. Que release was enhanced with an increase in the Que concentration. The initial release was higher, reaching 43–70% in 4 h. Then, a sustained release made it possible for it to be considered as a potential system for antitumor treatment. Antioxidant activity was not studied in this article. Polyhydroxylated fullerene (C_60_(OH)_n_) was loaded onto PLGA/PLC by Guo et al. [94] to obtain nanofibers for skincare purposes. The antioxidant capability of fullerene is well known. In this work, the ROS of HaCaT cells (keratinocyte cells from adult human skin) were examined to determine the antioxidant effects of fullerene. A protective effect was found at 2 h of incubation with peroxide. This was significantly related to untreated PLGA/PLC: when the concentration of fullerene increased, a higher effect was observed. Electrospun nanofibers of epigallocatechin-3-*O*-gallate (EGCG) loaded onto PLGA were prepared for antiadhesion applications by Shin et al. [95]. Given that EGCG has antioxidant properties, its antioxidant activity was assessed using a DCF assay. Pure PLGA and EGCG/PLA generated few ROS, and the ROS scavenging activities were significantly different from each other due to the presence of EGCG. ROS scavenging ability was found to be dependent on the quantity of EGCG. Llorens et al. [96] proposed the following fibers for applications in the purification of plasmidic or genomic DNA. Pyridoxal and pyridoxine are two forms of vitamin B_6_ that are water-soluble and necessary for hemoglobin synthesis and also for the metabolism of some proteins. They have an important role in DNA synthesis and have antioxidant properties. Caffeic and coumaric acids are among the hydroxycinnamic acids, which are considered to be valuable natural antioxidants for the phenolic groups in their composition. The effect of these compounds on oxidative DNA damage was tested individually (loaded onto PLA), aiming at purifying DNA subjected to oxidative stress. For pyridoxal and pyridoxine, fiber diameters decreased regarding unloaded PLA, while caffeic and coumaric acids affected fibers by decreasing their diameters compared to unloaded PLA. In addition, the inhibition of oxidative damage to plasmid DNA was satisfactory. Caffeic acid has been loaded onto both gelatin [21] and PLA polymers [96]. In terms of antioxidant activity, it showed remarkable activity loaded onto PLA against 2,2′ azobis (2-amidinopropane hydrochloride) (AAPH) that was comparable to coumaric acid and pyridoxine. Conversely, loaded onto gelatin, caffeic acid showed poor activity regarding phenolic compounds; thus, caffeic acid antioxidant activity depends on its interaction with the nanofiber surface. PLA and PVP were used by Miletić et al. [97] to load pomegranate seed oil (PSO), ethanolic extract of fermented pomegranate juice (EP), and cold-pressed sea-buckthorn oil (SB) (separately). Pomegranates (as berries) have been traditionally used due to their antioxidant and anti-inflammatory properties. On the other hand, sea buckthorn is often used in skincare products. Both PSO and SB have a significant content of oxidation-sensitive fatty acids, and PLA was the best polymer in preserving the antioxidant activity of compounds compared to PVP, which could have been due to the hygroscopic properties of PVP, since the presence of water in fibers might affect the oxidation process. According to a DPPH assay, the best fiber obtained was PLA containing 5% PSO and 5% EP, with possible applications in cosmetics. Chu et al. [98] reported the extraction of antioxidant essential oil from onions using a polystyrene nanofiber membrane. This was used instead of filter paper in a low-pressure filtration process. Onion juice was filtered, and the membrane was dried and eluted. Then, the solvent was distilled, obtaining the onion essential oil. The oil showed antioxidant activity in DPPH, ABTS, metal-chelating, and superoxide anion assays, while the antioxidant activity of this essential oil was dose-dependent. Finally, a nylon-6 nanofiber membrane was developed in coat form as a barrier to preserve the active surface of carbon electrodes. The effectiveness of the membranes was tested by measuring green tea to find out the antioxidant content. Signals obtained from a coated electrode were 80% clearer than those from an uncoated electrode [99].

## 4. Conclusions

Polymeric matrices containing antioxidant compounds as electrospun nanofibers were successfully reviewed. Regarding their high contact area, nanofibers containing compounds with antioxidant properties have been widely proposed for their application in the medical and food fields. Among polymers used as a matrix in the electrospinning process, synthetic polylactic acid and polycaprolactone have been the most widely used both individually and also as copolymers. In spite of natural polymers affording nanofibers improved characteristics such as biodegradability and/or antibacterial properties, they have scarcely been studied in terms of electrospun nanofibers. Nevertheless, silk fibroin, chitosan, cellulose, pullulan, hyaluronic acid, gliadin, PHBV, and zein have proven to be proper matrices for this purpose, both individually and as part of a whole system.

Various natural compounds and extracts have been identified as antioxidants. This property helps to inhibit free radicals and oxidative damage in tissues and foods. The most recurrent active compounds used are tannic acid, quercetin, curcumin, and vitamin E. The inclusion of active compounds in nanofibers often improves the bioavailability of these compounds, giving them increased stability, changing the mechanical properties of the polymers, enhancing nanofiber biocompatibility, and offering the required application new properties. The effects of active compounds on the morphologic properties of nanofibers are variable. Nanofiber diameter variation is mainly related to viscosity modification. Surface and softness have also been modified by including active compounds in different concentrations. Hydrophobicity or hydrophilicity could be enhanced by bonds and chemical interactions between compounds. The release of compounds from nanofibers is generally affected by concentration, release medium, pH, solubility, chemical matrix-active compound interactions, compound size, surface area, time, and/or temperature. Antioxidant activity depends mainly on the active compound concentration, matrix–radical interactions, compound release, and radical scavenging capability.

Polymer-free nanofibers are considered to have a high impact, as the encapsulation of active compounds allows for the active compound concentration and bioavailability to be increased without the need for additional polymers, enhancing the properties of nanofibers. Cyclodextrin has mostly been used as a matrix or carrier for active compounds due to its hydrophobic cavities, forming inclusion complexes that can be electrospun. Further research could focus on the utilization of natural polymers or free polymeric nanofiber production.

## Figures and Tables

**Table 1 nanomaterials-10-00175-t001:** Characteristics of nanofibers from eight reviewed articles with undefined applications.

Polymeric Matrix	Active Compound	Electrospinning Parameters	Average Diameter of Fibers (nm)	Reference
Polyvinylpyrrolidone (PVP)	Green tea extract	12.5 kV; 0.5 mL/h; 10 cm	344–386	[17]
PVP	*Garcinia mangostana* extract	15 kV; 0.5 mL/h	217–421	[18]
Poly(ethylene glycol) (PEG)	Tannic acid	17 kV; 20 cm	-	[19]
Pluronic and pullulan	Rutin	20 kV; 0.125 mL/h; 13 cm	100–102	[20]
Gelatin (GE)	Tannic, gallic, ferulic, and caffeic acids	25 kV; 0.3 mL/h; 13 cm	145–280	[21]
Polyvinyl alcohol (PVA)	Tannic acid/Fe^3+^ complexes	15 kV; 0.5 mL/h; 15 cm	144–337	[22]
Polyacrylonitrile (PAN)	Porphyrin	20 kV; 0.12 mL/h; 10 cm	Around 200	[23]
Poly(lactic acid) (PLA)	Rice extract	20 kV; 0.5 mL/h; 18 cm	450–656	[24]

**Table 2 nanomaterials-10-00175-t002:** Composition and characteristics of antioxidant nanofibers proposed for wound dressings from 16 reviewed articles.

Polymeric Matrix	Active Compound	Electrospinning Parameters	Diameter of Fibers (nm)	Fiber Characteristics	Reference
Chitosan–ethylenediaminetetraacetic acid/PVA	*Garcinia mangostana* extract (GME)	15 kV; 0.25 mL/h; 20 cm	205–251	Rapid GME release regarding matrix erosion	[30]
Silk fibroin (SF)	Grape seed extract (GSE)	14 kV; 0.3 mL/h; 15 cm	414–427	Antioxidant activity improved with the increase of GSE concentration	[12]
SF	Fenugreek seed extract	25 kV; 0.5 mL/h; 10 cm	438–640	The higher the fenugreek concentration, the more time-extended the inhibition of radical damage is	[31]
SF/hyaluronic acid (HA)	Olive leaf extract	20.46 kV; 0.12 or 0.36 mL/h; 10 cm	123	Antioxidant activity similar to original extract	[32]
Polycaprolactone (PCL)	Quercetin (Que)	16 kV; 0.6 mL/h; 10 cm	101	Antioxidant activity increases from 12% to 40% with incorporation of Que	[33]
PCL	*Clerodendrum phlomidis* leaf extract	12 kV; 12 cm	293	Antioxidant activity of resulting mats equivalent to *C. phlomidis* extract (between 68.4 and 69.4 μg/mL)	[34]
PCL/GE	Cerium oxide	1 kV/cm; 1 mL/h; 15 cm	300–760	PCL/GE/CeNPs nanofibers decrease by 30% the fluorescence intensity of DCF	[35]
PCL/PEG	Chrysin	18–22 kV; 2 mL/h; 20 cm	300–400	Fiber with the highest amount of chrysin maintained its antioxidant activity longer	[36]
PLA/polyethylene oxide (PEO)	Grape seed extract	17 kV; 1 mL/h; 9 cm	130–270	Near 85% 2,2-diphenyl-1-picrylhydrazyl (DPPH) scavenging even after 45 days	[37]
Poly(l-lactide-*co*-d,l-lactide) (coPLA)/PEG	Curcumin	17 kV; 3 mL/h; 10 cm	1360–1480	Curcumin reached 71–72% scavenging capacity	[38]
PLA	Curcumin	24 kV; 15 cm	333–380	DPPH inhibition reached 42–53% and decreased over time due to curcumin instability	[39]
Carboxymethylcellulose (CMC)/sodium alginate/CaCl_2_/PEO	Olive leaf extract	60 kV; 16 cm	167	89% of olive leaf extract radical scavenging	[40]
Cellulose acetate	Gallic acid (GA)	15, 18, or 21 kV; 1 mL/h; 15 cm	295–787	GA release was higher from the lowest concentrated fibers	[41]
PVA	Honey	20 kV; 0.3 mL/h; 10 cm	300–410	Antioxidant activity maintained after electrospinning process	[42]
Alginate/PVA	Honey	15 kV; 0.4 mL/h; 10 cm	378–528	DPPH scavenging resulted in 66% as the maximum obtained in 9 h	[43]
Polyurethane (PU)	Capsaicin	15 kV	150–500	DPPH radical scavenging of encapsulated capsaicin results were higher than capsaicin by itself (78% and 70%, respectively)	[44]

**Table 3 nanomaterials-10-00175-t003:** Characteristics of antioxidant nanofibers with proposed applications in tissue engineering from eight reviewed articles.

Polymeric Matrix	Active Compound	Electrospinning Parameters	Average Diameter of Fibers (nm)	Proposed Technology	Reference
SF	Melanin	1.5 kV/cm; 0.8 mL/h	343	Skeletal muscle tissue engineering (SMTE)	[47]
SF	Melanin	12 kV; 1 mL/h; 8 cm	800–840	Nerve regeneration	[48]
SF	Quinone-based chromenopyrazole (QCP)	15 kV; 0.5 mL/h; 10 cm	1400	Scaffolds for skin	[49]
PCL	Spirulina	15 kV; 15 cm	710	Scaffolds	[50]
PCL	Spirulina	12 kV; 0.3 mL/h; 15 cm	160–315	Scaffolds	[51]
PCL and poly(hydroxybutyrate-*co*-hydroxyvalerate) (PHBV)	Quercetin and curcumin	25 kV; 0.2 or 2 mL/h; 12 cm	332–556	Scaffolds	[52]
Cellulose	Tannic acid and phosvitin	17 kV; 20 cm	528–538	Scaffolds	[53]
Polylactide (PLA)	Vitamin B_6_, *p*-coumaric acid, and caffeic acid	17, 17.5, or 19 kV; 0.5 or 1 mL/h; 12.5 cm	81–101	Scaffolds	[54]

**Table 4 nanomaterials-10-00175-t004:** Characteristics of antioxidant nanofibers from nine reviewed articles and proposed applications in nanoencapsulation.

Polymeric Matrix	Active Compound	Electrospinning Parameters	Average Diameter of Fibers (nm)	Possible Application Areas	Reference
Zein	Quercetin/gamma-cyclodextrin inclusion complex	15 kV; 1 mL/h; 10 cm	750	Food and pharmaceutical	[58]
GE	Sour cherry concentrate	25 kV; 0.4 and 0.1 mL/h; 10 cm	-	Functional foods	[59]
PCL	α-tocopherol/β-cyclodextrin inclusion complex	15 kV; 0.5 mL/h; 8 cm	205–345	Topical drug delivery	[60]
Poly(amide-amine)/polyacrylic acid (PAA)/PVA	Thymol	23 kV; 0.8 mL/h; 23 cm	254–320	Food packaging	[61]
PLA	Gallic acid/cyclodextrin inclusion complex	15 kV; 1 mL/h; 10 cm	235–495	Food packaging	[62]
Poly (d,l-lactide-*co*-glycolide) (PLGA)/PEO	Ferulic acid	18 kV; 0.5 mL/h; 12 cm	150	Biomedical	[63]
PVA/PEO	β-carotene	10 kV; 0.1 mL/h; 10 cm	195–408	Industrial applications	[64]
Poly(2-hydroxyethyl methacrylate) (pHEMA)	Polyphenols	−2 and 10 kV; 0.23 mL/h; 15 cm	470–1930	Medical and biological	[65]
Various *	Curcumin	-	-	Wound healing	[66]

* Several polymers were reviewed in this article.

**Table 5 nanomaterials-10-00175-t005:** Characteristics of antioxidant nanofibers from 15 reviewed articles and proposed applications in food.

Polymeric Matrix	Active Compound	Electrospinning Parameters	Average Diameter of Fibers (nm)	Proposed Application	Reference
Gliadin	Curcumin	15 kV; 0.5 mL/h; 10 cm	375–410	Food Industry	[68]
Gliadin	Inclusion complexes of ferulic acid with hydroxypropyl-beta-cyclodextrins	18 kV; 1 mL/h; 10 cm	269–279	Food packaging	[69]
PHBV	Oregano essential oil, rosemary extract, and green tea extract	38 kV; 4 mL/h; 20 cm	800	Food packaging	[70]
Potato starch	Carvacrol	−3 and 25 kV; 0.6 mL/h; 20 cm	74–95	Food packaging	[71]
Chitosan/PVA	Fish-purified antioxidant peptide	15 kV; 0.2 mL/h; 15 cm	158–195	Food packaging	[72]
PLA/chitosan	Argan and clove oils			Food packaging	[73]
PVA/PAA/chitosan	Tannin acid/tripolyphosphate	Not reported (homemade)	132–578	Food packaging	[74]
GE	Curcumin	15 kV; 0.5 mL/h; 10 cm	295–368	Nutraceutical carrier	[75]
GE	Peppermint and chamomile essentials oils	15 kV; 0.3 mL/h; 10 cm	293–462	Edible food packaging	[76]
PCL	*Urtica dioica* L. extract	15 kV; 1 mL/h; 12 cm	575	Food preservation	[77]
PCL	Sage extract	19 kV; 3 mL/h; 15 cm	3300–3800	Food packaging	[78]
PLA	α-tocopherol/γ-cyclodextrin			Food packaging	[79]
PLA	Vitamin E	15 kV; 1 mL/h; 10 cm	140	Preservative packaging	[80]
PLA	Tea polyphenol	20 kV; 20 mL/h; 15 cm	490–680	Food packaging	[81]
PVA	Rosemary extract	30 kV; 2.2 mL/h; 20 cm	282	Hydrophilic and acid food products packaging	[82]

**Table 6 nanomaterials-10-00175-t006:** Nanofibers with antioxidant-related applications.

Polymeric Matrix	Active Compound	Electrospinning Parameters	Average Diameter of Fibers (nm)	Application Area	Reference
Cellulose acetate	Rosmarinic acid	20 kV; 0.25 mL/h; 12 cm	314–331	Transdermal patches	[91]
Cellulose acetate-based	Various *	-	-	Drug delivery	[92]
PLGA/PCL	Quercetin	25–28 kV; 0.1 mL/h; 12 cm	400–520	Implantable anticancer drug	[93]
PLGA/PCL	Polyhydroxylated fullerene	25 kV; 20 cm	200	Skin care	[94]
PLGA	Epigallocatechin-3-*O*-gallate	18 kV; 1 mL/h; 12 cm	300–500	Nanomedicine (postoperative adhesion prevention)	[95]
PLA	Vitamin B_6_, pyridoxal, and hydroxycinnamic acids	17–19 kV; 0.5–1 mL/h; 12.5 cm	81–101	Purification of DNA	[96]
PLA and PVP	Pomegranate seed oil (PSO), ethanolic extract of fermented pomegranate juice (EP), and cold-pressed sea-buckthorn oil (SB)	11–15.5 kV; 0.75–2.5 mL/h; 10–15 cm	820–1600	Cosmetics	[97]
Polystyrene (PS)	--	13.5 kV; 1.5 mL/h; 12 cm	300–600	Extraction	[98]
Nylon-6	--	24 kV; 0.03 mL/h	-	Electrochemistry	[99]

* Several active compounds were reviewed in this article.

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
