# Peer review of "Applications of Electrospun Nanofibers with Antioxidant Properties: A Review"

_nanomaterials, 2020, doi:10.3390/nano10010175_

Round 1

Reviewer 1 Report

Reviewer Comments:

This is a very interesting review related to the application of electrospun nanofibers with antioxidant properties.
Most of the paragraphs are well presented, however, there are a number of paragraphs that needs to be considered.

Comments:
-It will help the reader to have a small paragraph (probably after the Introduction) to review the different methods used for the evaluation of the antioxidant activity of the nanofibers.

-From the reviewed literature, there are also other compounds such as electrospun phospholipid fibers lipids that have been studied as antioxidant matrices; these studied can be also discussed at this review. For example: Shekarforoush, E. Mendes A.C., Beeren, S., Chronakis I.S. Electrospun Phospholipid Fibers as Micro-Encapsulation and Antioxidant Matrices. Molecules, 22, 10, (2017).

-The ‘antioxidant mechanisms’ of the electrospun nanofibers should be further elaborated and critically reviewed.

-Paragraph 3.3 Wound dressings
This paragraph is about Wound dressings, however, studies such as PVA and honey and PU foams for pillows are presented (lines 297 to 307). It is not clear why these studies are presented at this paragraph.

Conclusion, line 689: ‘Antidoxidant activity depended mainly on the morphological properties of the fibers’. This statement is not clearly documented at the manuscript. If this is the case, further clarification is need at the various paragraphs.

----End

Reviewer 2 Report

Please insert the tables in the text

In the whole text choose minute or min

Page 3 : If you put figure 1 in the text, you must also add table 1

Page 3 line 109 : there is a point of too much

Page 4 line 154 : plant in place of plants

Page 5 line 205 : sixteen in place of six-teen

Page 5 : you must also add table 2

Page 5 lines 214-219 : from silk fibroin to nanofibers of check the color of the letters

Page 6 line 226 : the number of the reference is 31 in place of 39

Page 7 line 281 increases in place of increase, line 290 there is too much space, line 319 table 3 not the same writing font and place table 3 after quoting reference 55

Page 8 line 320 reference 45 in place of 41

Page 8 line 328 was not studied in place of was no studied

Page 11 line 495 there is too much space

Page 12 line 539 there is too much space

Pge 12 line 562 Dadashpour et al. in place of Dadashpour

Page 13 lines 582 and 602 reported in place of reports

Page 14 line 627 there is too much space

Page 14 line 628 keratinocyte cells there in place of keratinocyte cell

Page 14 line 650 there is too much space

Page 14 line 663 write a in place of A

Page 26 table 2 do not write Polyurethane (PU) in bold

Reviewer 3 Report

In general, paper "Applications of electrospun nanofibers with antioxidant properties: A review" by A. Vilchez, F. Acevedo, M. Cea,  M. Seegerd and R. Naviad sounds interesting. The authors reviewed the current development of electrospun fibers with antioxidant properties. However, I do not like the form of presenting in the form  -

e.g., "112 publications belonging to various subjects areas were 83 found related to some application, two of which were discarded because they were written in Chinese 84 and Korean. All the articles reviewed were in English. The 75% of the 112 reviewed publications are 85 papers written between 2016 and 2018, lending support to the innovative nature of this review."

This is not a scientific approach to the subject, but only a form of reporting that should not have place in high-read magazines. I strongly advise to change this.

As well, I do not like Fig 1 that is trivial and adds nothing, I think that should be removed, changed or modified in a more advanced and modern way

I also advise to enreach all table with more detailed and important information e.g., diameter of fibers, composition, application, ect.. It will enable the redears to look through all revised papers and get more information about  the current state-of-knowledge in the filed of electrospun fibers with antioxidant properties.

Finally, I strongly recommend to add the recent papers related to this topic

Molecules, 2018, 23, 1399, "Evaluation of the Antimcrobial Activity and Cytotoxicity of ..."; Wound Healing Society, 2019, 8, 9, "Antimicrobial Electrospun Polycaprolactone-based Wound Dressings..."; European Polymer Journal, 2017,  96, 350–360 "Preparation and characterization of electrospun alginate nanofibers loaded with ciprofloxacin hydrochloride"

I recommand this paper to be published in Nanomaterials after minor changes.
